# Analysis of DNA methylation associates the cystine–glutamate antiporter *SLC7A11* with risk of Parkinson's disease

Costanza L. Vallerga[1], Futao Zhang[1], Javed Fowdar [2], Allan F. McRae [1], Ting Qi[1], Marta F. Nabais[1,3], Qian Zhang[1], Irfahan Kassam[1], Anjali K. Henders[1], Leanne Wallace[1], Grant Montgomery [1], Yu-Hsuan Chuang[4], Steve Horvath [5,6], Beate Ritz[4,7,8], Glenda Halliday [9], Ian Hickie[9], John B. Kwok[9,10], John Pearson[11], Toni Pitcher[12,13], Martin Kennedy [11], Steven R. Bentley[2], Peter A. Silburn[14], Jian Yang [1], Naomi R. Wray [1,14], Simon J.G. Lewis[9], Tim Anderson [12,13], John Dalrymple-Alford[12,15], George D. Mellick[2,17], Peter M. Visscher [1,14,17✉] & Jacob Gratten [1,16,17✉]

An improved understanding of etiological mechanisms in Parkinson's disease (PD) is urgently needed because the number of affected individuals is projected to increase rapidly as populations age. We present results from a blood-based methylome-wide association study of PD involving meta-analysis of 229 K CpG probes in 1,132 cases and 999 controls from two independent cohorts. We identify two previously unreported epigenome-wide significant associations with PD, including cg06690548 on chromosome 4. We demonstrate that cg06690548 hypermethylation in PD is associated with down-regulation of the *SLC7A11* gene and show this is consistent with an environmental exposure, as opposed to medications or genetic factors with effects on DNA methylation or gene expression. These findings are notable because *SLC7A11* codes for a cysteine-glutamate anti-porter regulating levels of the antioxidant glutathione, and it is a known target of the environmental neurotoxin β-methylamino-L-alanine (BMAA). Our study identifies the *SLC7A11* gene as a plausible biological target in PD.

[1] Institute for Molecular Bioscience, The University of Queensland, Brisbane, Australia. [2] Griffith Institute for Drug Discovery (GRIDD), Griffith University, Brisbane, Australia. [3] University of Exeter Medical School, Exeter EX2 5DW, Devon, UK. [4] Department of Epidemiology, Fielding School of Public Health, UCLA, Los Angeles, CA, USA. [5] Department of Human Genetics, David Geffen School of Medicine, University of California Los Angeles (UCLA), Los Angeles, CA, USA. [6] Department of Biostatistics, Fielding School of Public Health, UCLA, Los Angeles, CA, USA. [7] Department of Neurology, David Geffen School of Medicine, UCLA, Los Angeles, CA, USA. [8] Department of Environmental Health, Fielding School of Public Health, UCLA, Los Angeles, CA, USA. [9] Brain and Mind Centre & Faculty of Medicine and Health, The University of Sydney, Sydney, Australia. [10] School of Medical Sciences, University of New South Wales, Sydney, Australia. [11] Department of Pathology, University of Otago, Christchurch, New Zealand. [12] New Zealand Brain Research Institute, Christchurch, New Zealand. [13] Department of Medicine, University of Otago, Christchurch, New Zealand. [14] Queensland Brain Institute, The University of Queensland, Brisbane, Australia. [15] Department of Psychology, University of Canterbury, Christchurch, New Zealand. [16] Mater Research Institute, The University of Queensland, Brisbane, Australia. [17] These authors contributed equally: George D. Mellick, Peter M. Visscher, Jacob Gratten. ✉email: peter.visscher@uq.edu.au; jacob.gratten@mater.uq.edu.au

Parkinson's disease (PD) is a debilitating neurodegenerative disorder characterized by cytoplasmic and axonal aggregations of alpha-synuclein, known as Lewy bodies, and the progressive loss of dopaminergic neurons in the substantia nigra pars compacta (hereafter, substantia nigra) of the midbrain. The prevalence of PD is estimated to be ~1% in people over the age of 60, increasing to 3–4% at the age of 80[1]. With rising life expectancy worldwide, the number of individuals with PD is expected to more than double by 2040[2], leading to an increasingly heavy social and economic burden. The etiology of PD is complex, involving both genetic and environmental factors, but the specific molecular mechanisms contributing to pathogenesis remain poorly understood. Identifying epigenetic modifications associated with PD may provide insights into disease etiology.

DNA methylation at CpG dinucleotides is the most widely characterized epigenetic mechanism and is essential to normal development and maintenance of cell- and tissue-specific gene expression patterns[3]. Variation in DNA methylation can arise from environmental[4], stochastic[5], or genetic[6] perturbations, and there is growing evidence that DNA methylation could mediate the relationship between these processes in influencing risk of complex disease[7]. In the most recent epigenetic study of PD involving 289 PD cases and 219 controls[8], Chuang et al. reported 82 epigenome-wide significant CpG probes associated with disease, although none remained significant after accounting for differences in blood cell composition between cases and controls. Larger-scale studies of DNA methylation from PD case–control cohorts will be needed for identification of methylation probes robustly associated with disease risk, independent of cell type composition.

Here, we report results from analyses of genome-wide blood-based DNA methylation data in 1638 unrelated individuals (851 PD cases, 787 controls) of European descent from the System Genomics of Parkinson's Disease (SGPD) consortium (Supplementary Table 1) and 493 European individuals from the Parkinson's disease, Environment and Gene (PEG) cohort[9] (281 PD cases and 212 controls). We use the Houseman et al.[10] algorithm to impute blood cell count proportions in both the SGPD and PEG samples and find consistent differences in cell type composition between PD cases and controls. We show that a substantial proportion of the variance in PD status is captured by all methylation probes jointly and we use the PEG cohort to evaluate a DNA methylation-based classifier for PD based on the aggregated effect of all CpG probes in the SGPD discovery cohort. We perform methylome-wide association studies (MWAS) of PD in the SGPD and PEG data sets using recently developed mixed linear model-based MWAS methods[11], and we identify two previously unreported DNA methylation associations with PD in a meta-analysis of the SGPD and PEG MWAS summary data. We use Summary data-based Mendelian Randomization (SMR)[12] to demonstrate that one of these associations, with CpG probe cg06690548 in the SLC7A11 gene, is consistent with an environmental exposure, as opposed to genetic factors influencing DNA methylation or gene expression.

## Results

### Analysis of predicted cell type proportions.
We first used logistic regression to determine whether predicted cell type proportions (CTPs) were associated with PD status in the SGPD data set, after adjusting for sex and predicted age (Eq. (1), Methods). Compared with controls, we found that PD cases had more granulocytes (calculated as the sum of eosinophil and neutrophil proportions, $p < 2 \times 10^{-16}$), fewer B cells ($p = 2.16 \times 10^{-10}$) and fewer helper T cells (CD4T $p = 2.97 \times 10^{-15}$, CD8T $p = 5.29 \times 10^{-03}$), consistent with findings in a previous report[13] (Fig. 1). In addition, we observed a deficiency of natural killer cells (NK, $p = 7.53 \times 10^{-05}$) in PD cases compared with controls (Fig. 1). To investigate whether significant differences in CTPs between PD cases and controls could be owing to medication, the relationship between levodopa equivalent daily dosage (LEDD) and CTPs was investigated in 494 unrelated PD cases from the SGPD cohort for whom data on LEDD were available. We observed significant correlations (Bonferroni-adjusted $p$ value threshold $= 8.3 \times 10^{-03}$, correcting for testing of six cell types) between LEDD and the proportion of granulocytes (Pearson's correlation $= 0.17$, $p = 9.38 \times 10^{-05}$), CD4T cells (Pearson's correlation $= -0.14$, $p = 2.08 \times 10^{-03}$) and CD8T cells (Pearson's correlation $= -0.15$, $p = 1.08 \times 10^{-03}$; Supplementary Fig. 1), although none of these associations survived after adjusting for disease duration (Supplementary Fig. 2), which exhibited a strong and significant association with LEDD (Supplementary Fig. 3). Moreover, the case–control difference in CTPs was similar in analyses comparing controls with the most exposed cases (top 10% of LEDD), as to the least exposed cases (bottom 10% of LEDD) (Supplementary Table 2). These analyses suggest that medication exposure is unlikely to explain the CTP differences between PD cases and controls, although we are unable to exclude a small effect of PD medications on CTPs. For this reason, and because CTP strongly influences DNA methylation measures, we conservatively corrected for CTPs in our subsequent analyses.

### Phenotypic variance for PD attributable to all probes.
Next, we used the omics restricted maximum likelihood (OREML) method implemented in OSCA[11] to estimate the proportion of phenotypic variance for PD captured by all the probes ($\rho^2$) (Eqs. (2) and (3), Methods). For the SGPD and PEG cohorts, these estimates were 0.28 (se $= 0.05$, $p = 4.0 \times 10^{-20}$) and 0.24 (se $= 0.12$, $p = 1.4 \times 10^{-05}$) on the observed scale (i.e., 0–1), respectively, in models adjusted for sex, predicted age, predicted smoking exposure, and predicted CTPs.

### MWAS of PD in the SGPD data.
We then used OSCA to perform MOA (mixed linear model-based omic association) and MOMENT (multi component mixed linear model-based omic association excluding the target) testing of 263,264 CpG probes across 1638 unrelated PD cases ($N = 851$) and controls ($N = 787$) of European ancestry from the SGPD study (Eqs. (4) and (5), Methods). No inflation of the test statistics was observed in either MOA ($\lambda = 1.01$; Supplementary Fig. 4) or MOMENT ($\lambda = 1.04$; Supplementary Fig. 5), indicating that the mixed models efficiently corrected for potential confounding factors, consistent with simulations[11]. In the MOA analysis, we identified two epigenome-wide significant associations with PD (Bonferroni-adjusted significance threshold, $p < 1.9 \times 10^{-07}$; Fig. 2; Table 1), the strongest of which was for cg16001422 on chromosome 8 ($p = 2.3 \times 10^{-08}$, PLEC gene; Supplementary Fig. 6; Supplementary Data 1). Using the MOMENT method, which has been shown to be more robust and reliable than MOA for probes that are strongly associated with an unknown confounding factor[11], we did not identify any genome-wide significant associations (Supplementary Fig. 7; Supplementary Data 1).

### MWAS of PD in the PEG data.
Next, we performed mixed linear model-based association testing of 242,205 CpG probes in European samples (281 PD cases and 212 controls) from the PEG cohort (Methods), with no single CpG probe surpassing genome-wide significance in either the MOA or MOMENT analyses ($p < 2.06 \times 10^{-07}$, Fig. 2, Supplementary Figs. 4, 5, 7; Supplementary Data 2). Focusing on the MOA results, neither of the two epigenome-wide significant CpG's identified in the SGPD cohort replicated in PEG ($p_{rep} = 2.5 \times 10^{-02}$; Table 1), but 17 of the 21

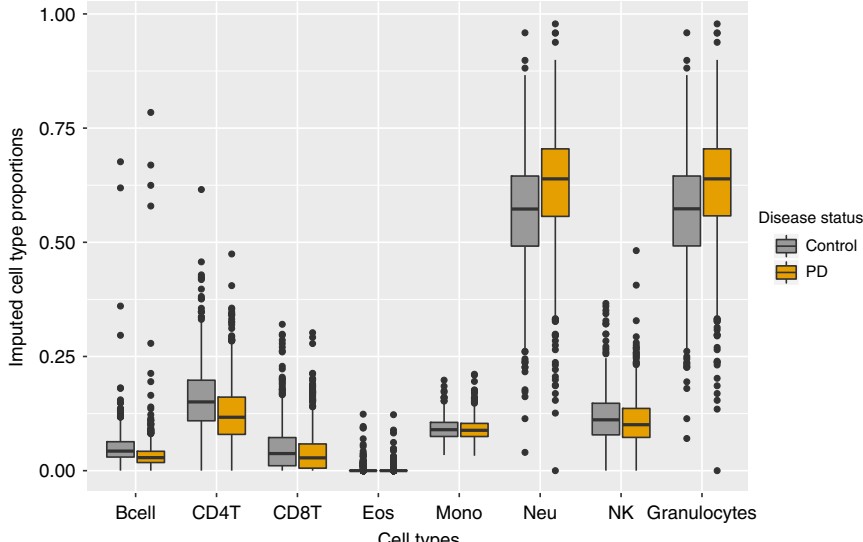

**Fig. 1 Distribution of predicted blood cell type proportions (CTPs) in PD and controls.** Boxplots of predicted blood CTPs in 1638 unrelated European individuals from the SGPD data set, stratified by PD status (851 PD cases, 787 controls). B cells, $p = 2.16 \times 10^{-10}$; CD4T cells, $p = 2.97 \times 10^{-15}$; CD8T cells, $p = 5.29 \times 10^{-03}$; eosinophils (eos) cells, $p = 1.43 \times 10^{-01}$; monocytes (mono) cells, $p = 2.22 \times 10^{-02}$; neutrophils (neu) cells, $p < 2 \times 10^{-16}$; natural killer (NK) cells, $p = 7.53 \times 10^{-05}$; granulocytes, $p < 2 \times 10^{-16}$. $P$ values from logistic regression of PD status on predicted age, sex, and individual cell types. Granulocytes are calculated as the sum of eosinophil and neutrophil proportions. Boxplot center lines show the median, box limits denote upper and lower quartiles, whiskers represent 1.5× interquartile range and individual points show outliers.

approximately independent (pairwise $R^2 < 0.1$) CpGs with $p < 1 \times 10^{-04}$ in the SGPD MWAS showed the same direction of effect in the PEG analysis (one-sided binomial test, $p = 3.6 \times 10^{-03}$; empirical $p \leq 6.2 \times 10^{-03}$ based on 10,000 random samples of 21 CpG probes; Supplementary Fig. 8). These observations are consistent with post hoc power analyses, which suggest that samples sizes in excess of 2000–3000 would be required for 80% power to replicate associations identified in SGPD (Supplementary Fig. 9). As expected, the effect sizes of CpG probes with $p < 5 \times 10^{-06}$ in the Chuang et al.[8] PEG analysis in which methylation was adjusted for blood CTPs were highly correlated with those in our analysis of the PEG cohort ($n = 10$ probes, Pearson's correlation = 0.93, $p = 8.9 \times 10^{-05}$, Supplementary Fig. 10), as were the effect sizes of genome-wide significant probes reported by Chuang et al. in their analysis without adjustment for blood CTPs ($N = 78$ shared probes, Pearson's correlation = 0.77, $p < 2.2 \times 10^{-16}$, Supplementary Fig. 11). However, none of the 78 shared probes was genome-wide significant in our analysis of the PEG data set, and nor did any single shared probe replicate in the SGPD data ($p < 0.05/78$).

**MWAS meta-analysis of PD.** We next conducted PD MWAS meta-analyses of MOA and MOMENT results for 229,071 CpG probes in 1132 PD cases and 999 controls from the SGPD and PEG cohorts (Methods). We identified two epigenome-wide significant probes in the MOA meta-analysis (Bonferroni-adjusted significance threshold, $p < 2.18 \times 10^{-07}$; Fig. 2; Table 2; Supplementary Fig. 4; Supplementary Data 3), including one not previously identified in the SGPD MWAS (cg06690548 on chromosome 4; Supplementary Fig. 12). In the MOMENT meta-analysis, no single probe was epigenome-wide significant (Supplementary Figs. 5, 7; Supplementary Data 3), but cg06690548 was the most strongly associated probe (beta = 0.92, se = 0.20, $p = 2.3 \times 10^{-06}$) and the effect sizes of CpGs with $p < 1 \times 10^{-04}$ in the MOA meta-analysis ($N = 58$) were highly correlated with those in the MOMENT meta-analysis (Pearson's correlation = 0.93, $p < 2.2 \times 10^{-16}$, Supplementary Fig. 13). Neither of the two epigenome-wide significant probes in the MOA meta-analysis

harbored a common SNP in the 3′ 5bp-subsequence of the probe that could have influenced our results. There was no evidence for strongly associated probes in known PD genes (e.g., *SNCA*) in either the MOA or MOMENT meta-analyses (Supplementary Data 4).

**Summary data-based Mendelian randomization.** Next, we used SMR[12] to identify specific genes associated with epigenome-wide significant methylation probes in the MOA analyses, and to evaluate evidence for genetic association between DNA methylation, gene expression and PD (Methods). Of the two significant probes in the MWAS meta-analysis based on MOA, and the additional epigenome-wide significant probe in the discovery MWAS, the only probe with a significant methylation quantitative trait locus (mQTL) ($p < 5 \times 10^{-08}$) in either blood or brain was cg06690548 on chromosome 4. SMR applied to this locus identified a significant association (Bonferroni-adjusted significance threshold correcting for 12 tests = $4.2 \times 10^{-03}$) between cg06690548 hypermethylation and downregulation of the neighboring gene *SLC7A11* (SMR: $p = 3.59 \times 10^{-03}$, Fig. 3 and Supplementary Table 3), but there was no evidence for a genetic association between PD and either cg06690548 methylation or *SLC7A11* expression (i.e., there was no evidence that the association of cg06690548 hypermethylation with PD was owing to genetic factors that influence cg06690548 methylation or *SLC7A11* expression). The HEterogeneity In Dependent Instrument (HEIDI) test[12] ($p = 0.16$) enabled us to rule out the possibility of an association between cg06690548 and *SLC7A11* expression owing to linkage disequilibrium between two distinct causal variants.

**DNA methylation-based classification of PD.** Finally, we examined the efficacy of a DNA methylation-based classifier for PD, using SGPD as the discovery sample and PEG as the target sample (Methods). We obtained best linear unbiased prediction-based estimates of effect sizes for 263,264 probes from a mixed linear model-based analysis in SGPD (adjusted for sex, predicted age, predicted smoking exposure, and predicted CTPs) and used these to generate methylation profile scores for each European

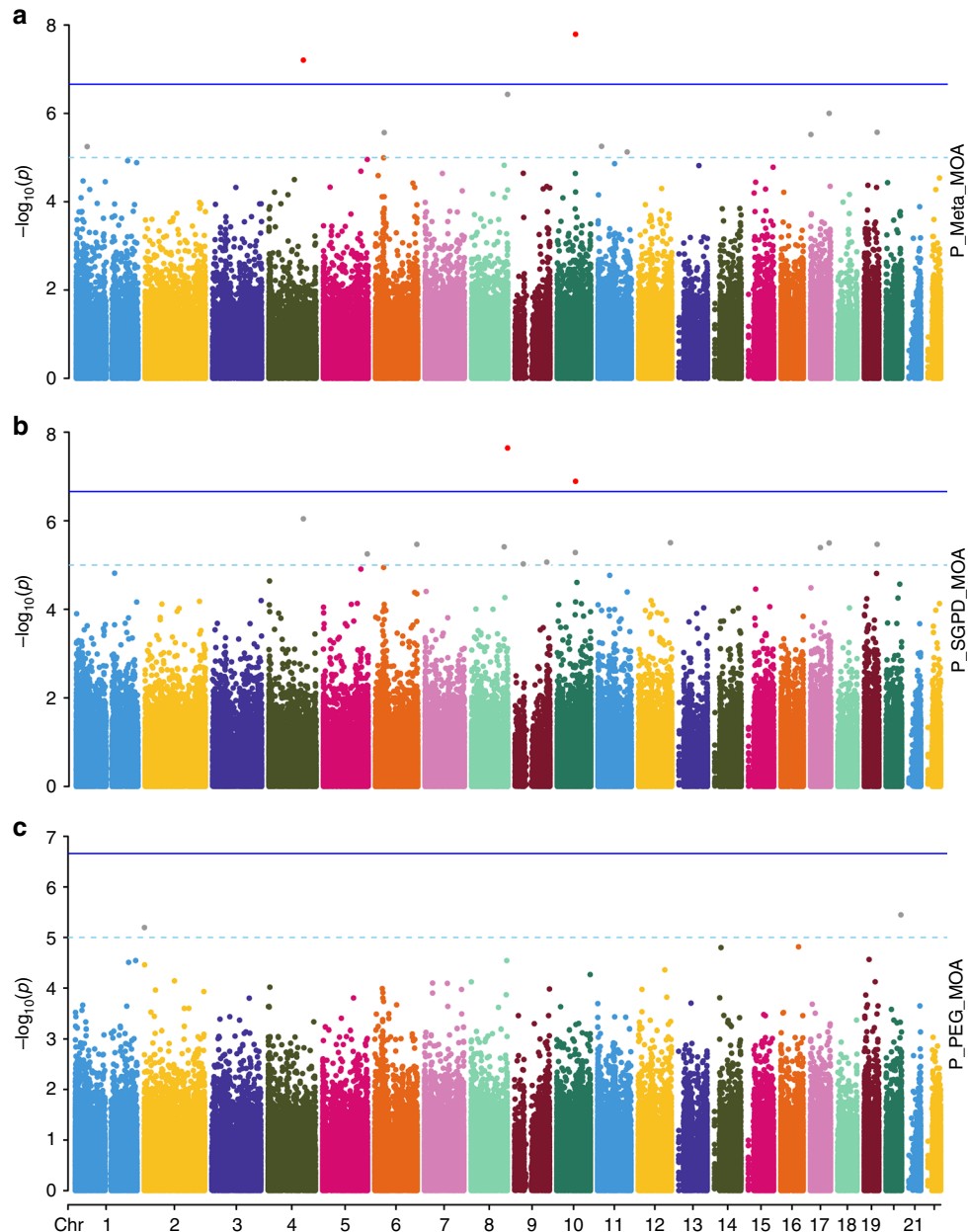

**Fig. 2 Manhattan plots of MOA MWAS of PD. a** MOA MWAS meta-analysis of PD in the SGPD and PEG cohorts ($N = 2131$ individuals); **b** MOA MWAS of PD in the SGPD data set ($N = 1638$ individuals); **c** MOA MWAS for PD in the PEG data set ($N = 493$ individuals).

| Chr | Probe | BP | Gene | Cohort | $b_{MOA}$ | $se_{MOA}$ | $p_{MOA}$ | $b_{MOM}$ | $se_{MOM}$ | $p_{MOM}$ |
|---|---|---|---|---|---|---|---|---|---|---|
| | | | | | | | | | | |
| 8 | cg16001422 | 145022842 | *PLEC* | SGPD | −3.05 | 0.54 | $2.3 \times 10^{-08}$ | −1.04 | 0.60 | $8.3 \times 10^{-02}$ |
| | | | | PEG | −0.44 | 0.96 | 0.65 | $-8.1 \times 10^{-03}$ | 0.91 | 0.99 |
| 10 | cg26033520 | 74004071 | *ASCC1* | SGPD | −1.55 | 0.29 | $1.3 \times 10^{-07}$ | −1.09 | 0.29 | $1.9 \times 10^{-04}$ |
| | | | | PEG | −1.17 | 0.56 | 0.04 | −0.77 | 0.53 | 0.15 |

**Table 1 Epigenome-wide significant probes for PD identified in SGPD.**

Results are shown for the two epigenome-wide significant probes in the MOA MWAS of SGPD, together with results for these probes from the MOMENT (MOM) MWAS of SGPD and the MOA and MOMENT MWAS's of the PEG replication data. Effect sizes (b) and standard errors (se) from OSCA are not standardized.

individual in the PEG sample. As expected, methylation profile scores were positively associated with PD in PEG (Fig. 4a), and in a logistic regression model adjusted for sex, predicted age, predicted smoking exposure and predicted CTPs, were found to capture 2.8% (Nagelkerke $R^2$, $p = 7.0 \times 10^{-04}$) of the variance in case–control status (on the observed scale). The area under the receiver operator characteristic curve (AUC) was 0.70 (95% CI = 0.66–0.75; Fig. 4b).

**Table 2 Epigenome-wide significant probes identified in the meta-analysis of SGPD and PEG.**

| Chr | Probe | BP | Gene | $b_{MOA}$ | $se_{MOA}$ | $p_{MOA}$ | $b_{MOM}$ | $se_{MOM}$ | $p_{MOM}$ |
|---|---|---|---|---|---|---|---|---|---|
| 4 | cg06690548 | 139162808 | *SLC7A11* | 1.09 | 0.20 | $6.2 \times 10^{-08}$ | 0.92 | 0.19 | $2.3 \times 10^{-06}$ |
| 10 | cg26033520 | 74004071 | *ASCC1* | −1.47 | 0.26 | $1.6 \times 10^{-08}$ | −1.01 | 0.25 | $7.5 \times 10^{-05}$ |

Results are shown for the two epigenome-wide significant probes identified in the MOA MWAS meta-analysis, together with results for these probes from the MOMENT (MOM) MWAS meta-analysis. Effect sizes (b) and standard errors (se) from OSCA are not standardized.

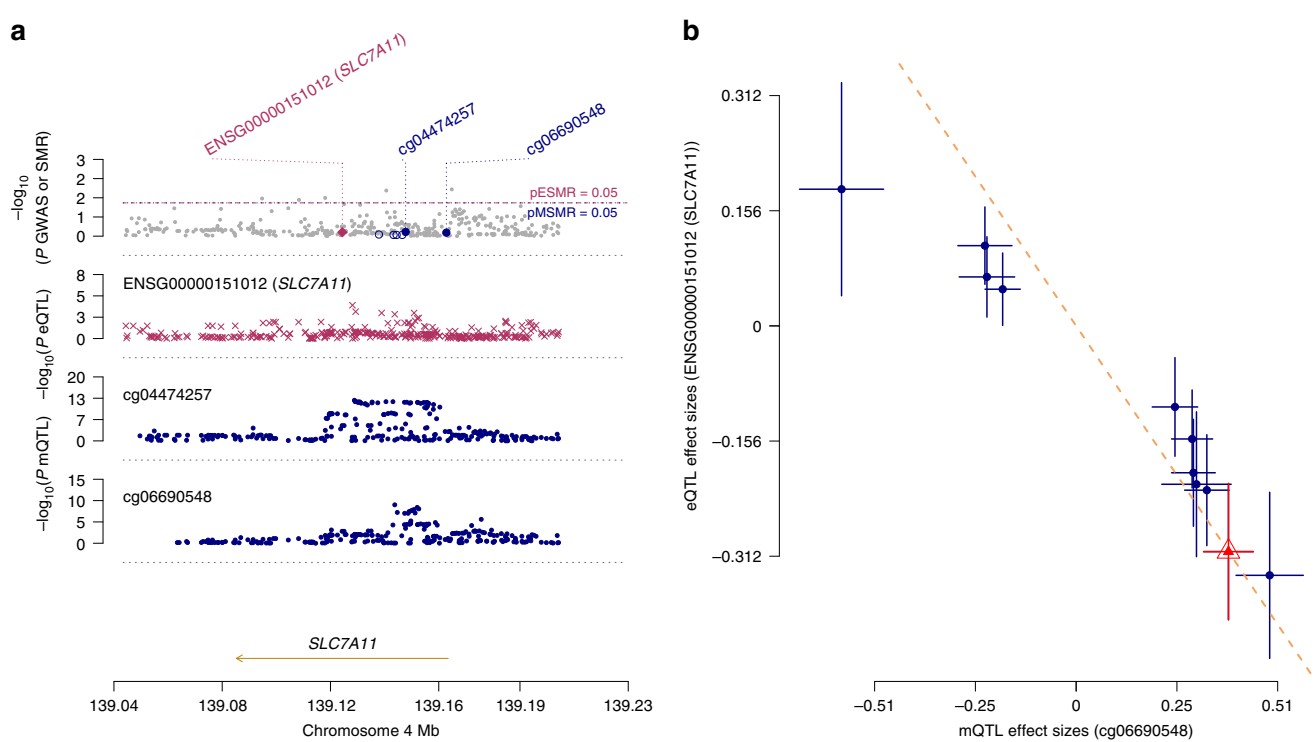

**Fig. 3 Summary data-based Mendelian randomization analyses at the *SLC7A11* locus. a** The uppermost plot shows −log$_{10}$(P values) of SNPs from the PDWBS GWAS meta-analysis[63]. The red diamonds and blue circles represent −log$_{10}$(P values) from SMR tests for association of gene expression and DNA methylation probes with PD, respectively. Neither *SLC7A11* nor cg06690548 show evidence for a genetic association with PD. The middle plot shows −log$_{10}$(P values) of the SNP associations for gene expression probe ENSG00000151012 (tagging *SLC7A11*) from the Brain-eMeta eQTL study[57]. The bottom plot shows −log$_{10}$(P values) of the SNP associations for DNA methylation probe cg06690548 from the Brain-mMeta mQTL study[57]. **b** Relationship between effect sizes of mQTLs for cg06690548 from the Brain-mMeta mQTL study[57] and the corresponding effect sizes for gene expression probe ENSG00000151012 from the Brain-eMeta eQTL study[57] (SMR, $p = 3.59 \times 10^{-03}$). The red triangle shows the top cis-mQTL, blue circles indicate cis-mQTLs. Error bars show the standard errors of the SNP effects.

## Discussion

In our analysis of blood-based DNA methylation in 1132 PD cases and 999 controls, we identified two DNA methylation probes associated with PD using MOA: cg26033520 on chromosome 10 in the vicinity of the *ASCC1* (Activating Signal Co-integrator 1 Complex subunit 1) gene and cg06690548 on chromosome 4 in the promoter of the *SLC7A11* (Solute Carrier Family 7 member 11) gene. The latter was also the most strongly associated probe in the MOMENT meta-analysis, suggesting that it is unlikely to be a false positive, despite not surpassing epigenome-wide significance using that method. Our results are consistent with previous simulations showing that MOMENT is more reliable and robust but slightly less powerful than MOA[11].

A particularly interesting finding from the SMR analysis was that hypermethylation of cg06690548 in PD was associated with downregulation of *SLC7A11* but there was no evidence for a genetic association between PD and either cg06690548 methylation or *SLC7A11* expression. To our knowledge the SMR method has not previously been used to rule out a genetic effect

underlying a disease association. The SMR results imply that the association of cg06690548 with PD is not owing to genetic factors and so may reflect a PD-related environmental exposure or be a consequence of disease (e.g., a medication effect). We investigated the latter possibility by applying MOA (with sex, predicted age, predicted smoking exposure, and predicted CTPs as covariates) to data on LEDD in 494 unrelated European PD cases in the SGPD cohort. We found no evidence for an association of cg06690548 methylation with PD medication dosage (beta = 0.50; s.e. = 0.37, $p = 0.17$; unpublished data), although we cannot rule out the possibility of a brain-specific association.

The *SLC7A11* gene encodes a sodium-independent cysteine-glutamate antiporter known as system Xc- (or xCT). The antiporter couples the release of one molecule of intracellular glutamate, which is necessary for excitatory signaling between neurons, with the uptake of one molecule of extracellular cystine, which is rate limiting for synthesis of glutathione, the primary antioxidant in the brain. Downregulation of system Xc- results in decreased intracellular levels of glutathione[14,15], which in

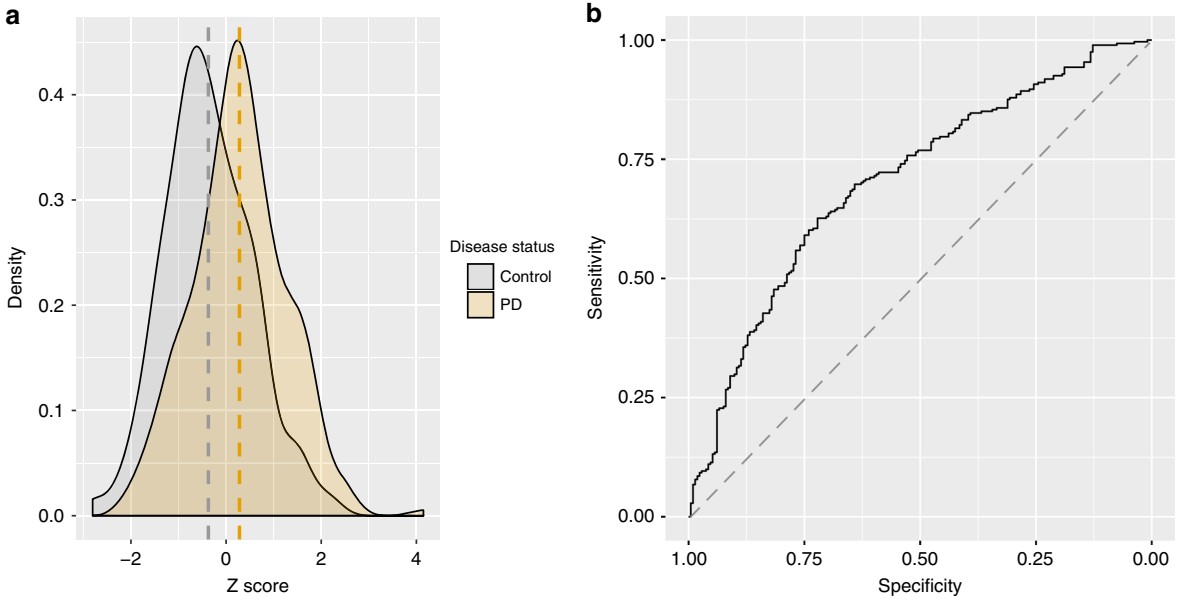

**Fig. 4 DNA methylation-based classification of PD. a** Density distribution of the SGPD-derived DNA methylation-based classifier in the PEG cohort. Orange and gray represent the distributions of the z-scaled classifier in PD cases and controls in the PEG cohort (respectively). **b** Area under the receiver operator characteristic curve (AUC) of the SGPD-based DNA methylation classifier in the PEG cohort (AUC = 0.70, 95% CI = 0.66–0.75).

turn leads to increased oxidative stress. Disruption of glutamate signaling has been associated with multiple neurodegenerative diseases[16], and for reasons that are not yet understood, dopaminergic neurons in the substantia nigra appear to be especially susceptible to damage by reactive oxygen species[17]. This is notable given that loss of dopaminergic neurons in the substantia nigra is the hallmark pathology of PD and reduced levels of glutathione have been reported in the substantia nigra, but not other brain regions, and in human olfactory neural stem cells in PD patients compared with aged-matched controls[15,18,19]. In this context, our SMR findings suggest that the mechanism underlying the association of cg06690548 hypermethylation with PD is that this causes downregulation of system Xc- (Supplementary Table 3), resulting in reduced glutathione levels and increased oxidative stress, thereby triggering degeneration of dopaminergic neurons in the substantia nigra.

Intriguingly, system Xc-, coded by the *SLC7A11* gene, is a target of the environmental neurotoxin β-methylamino-ʟ-alanine (BMAA)[20], a non-protein amino acid produced by cyanobacteria (i.e., blue green algae)[21]. Chronic dietary exposure to BMAA is believed to be the cause of the amyotrophic lateral sclerosis/parkinsonism-dementia complex (ALS/PDC) that in the 1950′s had a 50 to 100-fold higher prevalence in the Chamorro people of Guam than in developed countries[22–25]. Support for the BMAA hypothesis in neurodegeneration has waxed and waned in the 30 years since the hypothesis was first proposed[26–30], but there has been renewed interest in recent years based on diverse evidence, including that dietary exposure to BMAA can induce neurofibrillary tangles and β-amyloid plaques similar to those in brain tissues from Chamorros with ALS/PDC[31] and that regional variation in ALS prevalence is associated with cyanobacterial algal blooms[32]. The mechanism of BMAA neurotoxicity is thought to involve competition between BMAA and cystine at the cystine/glutamate antiporter, leading to reduced uptake of cystine, a depletion of intracellular glutathione and increased oxidative stress[33]. Concurrently, this competition may also lead to excitotoxicity via elevated glutamate release and activation of metabotropic glutamate receptor 5 (mGluR5)[34], and a third possibility is that BMAA transported by system Xc- may be mis-incorporated

in human proteins in place of l-serine, leading to misfolding and aggregation[35].

Our findings, although not directly implicating BMAA in risk of PD, are notable for being entirely independent of previous reports—spanning several decades—for an association of this molecule with neurodegenerative disease. More research is warranted to determine whether dietary exposure to BMAA results in hypermethylation of cg06690548, if altered methylation at this CpG also occurs in the substantia nigra, and whether these methylation changes are associated with an increase in oxidative stress in these cells via downregulation of system Xc-. Confirmation of a causal link between BMAA exposure and risk of PD would have significant public health implications, as monitoring and control of BMAA in the food chain may be an efficient and cost-effective intervention for Parkinson's disease and other neurodegenerative disorders. We acknowledge that environmental factors other than BMAA, such as pesticide exposure, could also plausibly influence cg06690548 methylation and *SLC7A11* expression in PD.

The gene closest to the most significant probe (cg26033520) identified in our MOA MWAS meta-analysis (ranked 35th out of >219 K probes in the MOMENT meta-analysis) was *ASSC1*, which encodes a subunit of the activating signal co-integrator 1 complex. *ASCC1* is a compelling candidate for PD because a recessive mutation in this gene has been associated with spinal muscular atrophy with congenital bone fractures-2 (SMABF2)[36], and studies in animal models have demonstrated that *ASCC1* knockdown is associated with impaired axonal outgrowth of alpha-motor neurons, impaired neuromuscular junction formation and compromised motor functioning[36]. More generally, *ASCC1* is a transcriptional activator that has an important role in gene transactivation via multiple transcription factors including nuclear factor kappa-B (NF-kB). NF-kB has a key role in regulating the immune response to infection and dysregulated NF-κB activation has been associated with cancer, inflammatory, and autoimmune diseases and septic shock[37]. This is noteworthy given recent findings from in vivo studies of PD, supporting a role for immune system dysfunction in disease onset and progression, including via cytokine activation and neuroinflammation[38].

The strongest signal in the SGPD MOA MWAS was for cg16001422, located in the proximity of the *PLEC* gene. This probe failed to replicate in MOA analysis of PEG, did not achieve epigenome-wide significance in the MOA meta-analysis and was not supported by MOMENT in SGPD, PEG or the meta-analysis. Nonetheless, it is noteworthy that *PLEC*, which encodes a large polypeptide acting as a crosslinker between actin microfilaments, microtubules, and intermediate filaments in the cell cytoskeleton[39], has previously been associated with multiple autosomal recessive forms of muscular dystrophy, including epidermolysis bullosa simplex with muscular dystrophy, and muscular dystrophy, limb-girdle, autosomal recessive 17[40–43]. Our results suggest that further investigations are merited on the role of *PLEC* in risk of PD.

The proportion of variance in PD status associated with all the methylation probes ($\rho^2$) was consistently high in both the SGPD and PEG cohorts (0.28 and 0.24, respectively), and similar in magnitude to the variance in PD status explained by common SNPs (0.28 in SGPD, estimated using the genomic-relatedness-based restricted maximum likelihood method implemented in the GCTA software[44]). Notably, when the two epigenome-wide significant probes identified in the MOA SGPD MWAS were fitted as fixed effects (in addition to sex, predicted age, predicted smoking exposure, and predicted CTPs) in an OREML analysis of the PEG data, $\rho^2$ decreased from 0.24 to 0.21. These results imply that the majority of variance in PD status associated with DNA methylation is still unaccounted for, and that larger PD MWAS are likely to identify additional genomic regions associated with PD.

A DNA methylation classifier based on best linear unbiased prediction-based effect sizes for all probes from an mixed linear model analysis of SGPD captured ~2.8% of the variance in PD case–control status in the PEG data set. This estimate was highly statistically significant but lower than previously reported adjusted $R^2$ estimates for methylation-based classifiers of other complex traits (e.g., BMI, $R^2 = 12.2\%$ in the Lothian Birth Cohorts[45,46]). The receiver operator characteristic curve is used in clinical epidemiology to quantify how well medical diagnostic tests discriminate between affected and non-affected (or controls) individuals. Accuracy is measured by the area under the receiver operator characteristic curve (AUC). An AUC of 1 represents a perfect test, whereas an AUC of 0.5 represents a test with no discriminatory power. In this study, the AUC of the methylation-based classifier for PD was estimated at ~ 0.70. This value is comparable to that of the latest GWAS-based predictors for PD[47], but we do not know if PD-associated methylation differences are a cause or consequence of the disease. For instance, methylation differences may be owing to medication effects or to some other physiological change that occurs as a consequence of having PD. This has implications for the utility of methylation-based classifiers, as they may differentiate PD cases from controls, but have no predictive value in terms of identifying individuals at high risk of PD in the population (i.e., before they develop the disease).

Finally, prior studies of PD[13] have reported disease-associated differences in blood CTPs. We replicated a number of these CTP differences in analysis of PD cases and controls in our SGPD data, but nonetheless chose to adjust for CTPs in our MWAS analyses, owing to the fact that we could not rule out the possibility that PD medications contributed to these differences. We acknowledge that this approach is conservative and that true positive methylation associations with PD may have been overlooked as a result. Further investigations will be needed to establish the extent to which CTP differences associated with PD are part of the disease process, as opposed to a consequence of the disease. We also acknowledge that analysis of whole blood-based DNA is a limitation, given that PD is a disorder of the central and peripheral nervous system. Analysis of DNA methylation in post-mortem brain tissues from PD cases and controls should be a future priority for the field.

In conclusion, we describe two previously unreported associations between DNA methylation and PD, including with the *ASCC1* gene on chromosome 10 and the *SLC7A11* gene on chromosome 4. These associations provide support for glutamate signaling disruption, oxidative stress, and neuroinflammation as potential etiological mechanisms contributing to PD. The *SLC7A11* association is especially notable given prior evidence that this gene is a target of the neurotoxin BMAA. An important caveat of our study is that epigenome-wide significant associations were only identified with MOA, which has a higher false positive rate than MOMENT[11]. However, we have shown that the two probes achieving epigenome-wide significance in the MOA meta-analysis were also among the most strongly associated probes in the MOMENT meta-analysis (ranks 1 and 35 for cg06690548 and cg26033520, respectively). This suggests that these associations are unlikely to be false positives, and that their failure to achieve epigenome-wide significance in MOMENT is likely owing to the slightly reduced power of this method relative to MOA[11]. Our linear mixed model analyses revealed that a surprisingly high proportion of variance in PD status was associated with DNA methylation measured in whole blood, suggesting that studies of DNA methylation in larger PD case–control cohorts are likely to yield additional discoveries relevant to the disorder. A DNA methylation classifier based on the SGPD data captured a significant proportion of the variance in PD status in the independent PEG cohort, but an improved understanding of the mechanisms of cause and effect in relation to DNA methylation probes altered in PD is required prior to the application of DNA methylation-based classifiers as a diagnostic tool.

## Methods

**Study populations**. The System Genomics of Parkinson's Disease (SGPD) cohort comprises genotype, phenotype, and DNA methylation data for a total of 2333 participants (1292 PD cases, 1041 controls) recruited from three different studies across Australia and New Zealand: (1) the Queensland Parkinson's Project (QPP), (2) the New Zealand Brain Research Institute PD case–control cohort (NZBRI), and (3) the Sydney PD case–control cohort (SYD).

The QPP cohort includes 1791 participants (867 PD cases, 924 controls) mostly of European ancestry. PD was diagnosed according to standard criteria[48] and controls consisted of healthy community-based, age-matched volunteers residing in the same area and from the same ethnic background as the PD patients ($N = 507$), together with patients' spouses ($N = 266$) and siblings ($N = 151$). Whole blood or saliva ($N = 4$) samples were collected at the time of recruitment. A total of 1692 individuals completed the Parkinsonism and related neurological disorders survey and underwent the same evaluations, which included questionnaires on demographics, medical history, environmental exposures and the Geriatric Depression Scale (GDS). In addition, cases undertook a cognition test summarized by the Unified Parkinson's Disease Rating Scale (UPDRS) score.

The NZBRI cohort comprises 210 participants (151 PD cases, 59 matched controls) recruited by the NZBRI. Exclusion criteria for PD patients were prior history of learning disability, severe head injury, stroke, or other neurological impairment and major psychiatric complications at the point of study entry. Whole blood samples were collected at the same time as phenotypic measurements, which included demographic, medical, and environmental exposure information for all participants. In addition, cases underwent periodic cognition tests, which included the Parkinson's Disease Dementia (PDD) criteria[49], the Parkinson's Disease and Mild Cognitive Impairment (PD-MCI) criteria[50], and the UPDRS score.

In the SYD cohort, 332 participants (274 PD cases, 57 matched controls, 1 individual with missing phenotype) were recruited from the Parkinson's Disease Research Clinic, Brain and Mind Research Institute at the University of Sydney. PD was diagnosed according to the UKPDS Brain Bank clinical diagnostic criteria[51].

The PEG study is a large population-based study of mostly rural and township residents of California's central valley[9]. The PEG study comprises of 508 European (289 PD cases, 219 controls) and 64 Hispanic individuals (46 PD cases, 18 controls) for a total of 334 PD cases and 237 controls. Cohort details and DNA extraction methods are described elsewhere[9,13].

**Ethics approval and consent to participate**. All participants gave written consent. The QPP study was approved by the Griffith University Human Research Ethics Committee (ESK0411HREC). The Southern Health and Disability Ethics committee (New Zealand) approved the NZBRI study protocol (URA/11/08/042; URB/

09/08/037). The SYD study protocol was approved by the University of Sydney Human Research Ethics Committee (10963; 2013/945). Approval for analysis of SGPD samples was granted by the University of Queensland Human Research Ethics Committee (2011001173). The PEG study was approved by the UCLA Institutional Review Board (IRB 11–001530).

**Genotyping and quality control**. DNA samples from SGPD participants and 26 plate controls were genotyped using the Illumina PsychArray-B.bpm (571,054 SNPs) at the QIMR Berghofer Molecular Epidemiology Laboratory. Standard individual-level and SNP-level quality control was applied to the data. A total of 39 unique samples were excluded from the SGPD data set on the basis of unresolved sex discrepancies (20 samples), high missing data rate (genotype failure rate 0.03; 17 samples) and sample duplications ($PI_{hat} > 0.9$; 10 samples). SNP markers were excluded if they had missing genotype rate >5%, Hardy–Weinberg Equilibrium (HWE) test $p$ value $< 1 \times 10^{-05}$ in controls, different missing genotype rates between cases and controls ($p$ value $< 1 \times 10^{-05}$) or low minor allele frequency (MAF < 0.01). We additionally dropped a total of 5293 duplicated SNPs. This screen reduced the total number of analyzed SNPs by ~49% (predominantly owing to the MAF filter), leaving 288,452 SNPs and 2227 individuals (1688 from QPP, 210 from NZBRI, 329 from SYD). We performed multidimensional scaling analysis on the cleaned SNP data to establish the genetic ancestry of study participants. We merged the SGPD sample with the HapMap3 data set comprising 988 individuals across 11 populations and defined Europeans as those falling within ±5 SD of the mean of the HapMap3 European cluster based on the top two principal components. A total of 78 individuals (62 from QPP, 4 from NZBRI and 12 from SYD) showed evidence for non-European genetic ancestry and were removed. The cleaned SGPD genotypes were imputed to the Haplotype Reference Consortium[52] and then additional filtering was performed to remove SNPs with low MAF < 0.01, HWE test $p$ value in controls $< 1 \times 10^{-05}$ and INFO score <0.3. This screen reduced the total number of imputed SNPs (~38 million) by 81%, leaving 7,582,086 SNPs and 2138 individuals of European ancestry (1169 PD cases, 65% male; and 968 controls, 46% male, 1 individual with missing phenotype). These data were used solely for the purpose of identifying unrelated individuals for inclusion in DNA methylation analyses, as described below.

**DNA methylation and quality control**. Whole blood-derived genomic DNA from a total of 1974 SGPD individuals was bisulphite-converted using the EZ-96 DNA Methylation kit (Zymo Research). The Human Methylation 450 K BeadChip was used to assess methylation status at 485,512 CpG sites across the genome. Data were available on 1704 (820 cases, 884 controls) QPP participants, 210 (151 cases, 59 controls) NZBRI participants, and 60 (30 cases, 30 controls) SYD participants. Methylation arrays were run in two batches: the first comprising 940 samples (835 QPP, 105 NZBRI) and the second comprising 1034 samples (869 QPP, 105 NZBRI, 60 SYD). Samples were randomly placed with respect to array and array position in order to minimize the potential for batch effects. Low-quality probes and samples were excluded from further analysis as described below. Methylation scores for each CpG site, obtained as a ratio of the intensities of fluorescent signals, are represented as β-values, which range between 0 and 1; a value of 0 indicates that all copies of the CpG site in the sample (i.e., all cells) were completely un-methylated, and a value of 1 indicates that every copy of the site was methylated. Raw intensity data were background-corrected and normalized using internal controls, and methylation β-values were generated using the R meffil package. Quality control (QC) was performed to remove probes with a low (<95%) detection rate at $p < 0.01$ and those that failed the minimum threshold for the number of beads ($N = 3$). The R meffil package was also used to perform sample QC using Illumina recommended thresholds. Samples were dropped if call rate was low (<450,000 probes detected at $p < 0.001$), if predicted sex, based on XY probes, did not match reported sex, and if predicted median methylated signal was more than three standard deviations from the expected. A total of four individuals whose DNA was extracted from saliva were also excluded. After these QC steps, methylation β-values were quantile-normalized with respect to 20 principal components (PCs) generated from the control matrix and the most variable probes. In addition, normalization was adjusted for batch, slide, cohort, sentrix row/column, sex, and age. A total of 485,237 probes and 1889 samples (959 PD cases, 930 controls) were retained following QC. We used the GCTA software[44] to generate a genetic relationship matrix (GRM) using all ~7.5 M imputed QC-pass SNPs, and removed one of each pair of individuals (total $N = 190$) with an estimated pairwise genetic relationship >0.05. Subsequently, the Omics-data-based Complex trait Analysis (OSCA) software[11] (see http://cnsgenomics.com/software/osca) was used to perform additional filtering of cross-reactive probes (29,218), X/Y-chromosomes probes (10,711) and lowly variable probes (sd < 0.02), leaving 263,264 probes and 1638 unrelated European individuals (851 PD cases, 787 controls) available for analysis.

We downloaded Illumina Infinium Human Methylation 450 K BeadChip data from 508 European (289 PD cases, 219 controls) and 64 Hispanic individuals (46 PD cases, 18 controls) in the PEG study from GEO (https://www.ncbi.nlm.nih.gov/geo/query/acc.cgi?acc=GSE111629), with the purpose of replicating epigenome-wide significant probes identified in the SGPD EWAS and performing meta-analysis. We performed DNA methylation QC on the European samples from the PEG data using the same R meffil pipeline applied to the SGPD data set. Overall, 15 unique European samples were filtered out owing to low call rate (<450,000 probes

detected at $p < 0.001$; $N = 10$) or because predicted sex did not match reported sex ($N = 6$). After these QC steps, methylation β-values were quantile-normalized with respect to 15 PCs generated from the control matrix and the most variable probes. In this study, we chose 15 PCs as opposed to 20 because the variance captured by PCs 16–20 was minimal. In addition, normalization was adjusted for slide, sentrix row/column, sex, and age. Finally, cross-reactive probes, X/Y-chromosomes probes and lowly variable probes (sd < 0.02) were also filtered out leaving 242,205 probes and 493 European samples available for analysis.

**Cell type proportions**. The R Meffil package was used to impute blood CTPs according to the Houseman et al.[10] method for each individual in both the SGPD and the PEG cohorts. Differences in CTPs between PD cases and controls were then singularly tested through logistic regression of PD status on covariates (predicted age, sex) to account for possible confounding factors. Specifically, we used the following model:

$$\mathbf{y} = \beta_0 + \sum_i \beta_i \mathbf{X}_i + \mathbf{e} \qquad (1)$$

where $\mathbf{y}$ is an $n \times 1$ vector of $n$ individuals representing the log odds that an individual is a case, $i \in [1, k]$ with $k$ being the number of fitted covariates, $\beta_i$ is the effect of the $i$th covariate on the phenotype, $\mathbf{X}_i$ is the $i$th independent variable or covariate and $\mathbf{e}$ is a $n \times 1$ vector of residuals. Finally, individual CTPs in 494 unrelated PD cases from the SGPD cohort were regressed on LEDD calculated accordingly to Tomlinson et al.[53]. LEDD is a quantitative measure of PD-specific medication measured as mg/day. In a few instances, exact LEDD was not available but was reported as a range (e.g., 530–1800 mg); in these cases, LEDD was calculated as the mid-point between the lowest and highest values in the range. Prior to the regression analysis, LEDD was log-transformed owing to the right skewness of its distribution and standardized.

**OREML analyses**. The OREML method implemented in the OSCA[11] software is a model in which a single random effect component is used to estimate the proportion of variance in a specific trait captured by all the DNA methylation probes. The model can be written as:

$$\mathbf{y} = \mathbf{C}\beta + \mathbf{W}\mathbf{u} + \mathbf{e} \qquad (2)$$

where $\mathbf{y}$ is an $n \times 1$ vector of phenotype values of $n$ individuals (coded as 1 for PD and 0 for unaffected), $\mathbf{C}$ is an $n \times p$ matrix for covariates (e.g., sex, age) with $p$ being the number of fitted covariates, $\beta$ is a $p \times 1$ vector of the effects of the fixed covariates on the phenotype, $\mathbf{W}$ is an $n \times m$ matrix of $m$ standardized DNA methylation values, where $m$ is the number of DNA methylation sites, $\mathbf{u}$ is an $m \times 1$ vector of the joint effects of all the probes on the phenotype, and $\mathbf{e}$ is an $n \times 1$ vector of residuals. The variance-covariance matrix for $\mathbf{y}$ can be written as:

$$\mathbf{var}(\mathbf{y}) = \mathbf{V} = \mathbf{W}\mathbf{W}'\sigma_u^2 + \mathbf{I}\sigma_e^2 = \mathbf{A}\sigma_o^2 + \mathbf{I}\sigma_e^2 \qquad (3)$$

where $\mathbf{A}$ (the omics-data-based relationship matrix or ORM) is defined as $\mathbf{W}\mathbf{W}'/m$ and $\sigma_o^2 = m\sigma_u^2$. In such a mixed linear model, $\sigma_o^2$ and $\sigma_e^2$ are the variance components and the amount of variance in the phenotype attributable to all the probes is defined as $\rho^2 = \sigma_o^2/(\sigma_o^2 + \sigma_e^2)$. Estimation of the variance components can be computed using REML[54], which in OSCA nomenclature is OREML. In the context of DNA methylation data, $\rho^2$ parallels the concept of SNP-based heritability as implemented in GCTA[44], with the caveat that association signals of DNA methylation data may reflect both cause and consequence of disease. Here, estimates of $\rho^2$ were from a model that included both confounder covariates (sex, predicted age, predicted smoking exposure) and predicted CTPs (with the exclusion of eosinophils).

**Mixed linear model-based MWAS of PD**. We used two mixed linear model-based approaches implemented in the OSCA[11] software to test for association between disease status and DNA methylation: MOA and MOMENT. Specifically, the MOA model is:

$$\mathbf{y} = \mathbf{w}_i b_i + \mathbf{C}\beta + \mathbf{W}\mathbf{u} + \mathbf{e} \qquad (4)$$

where $\mathbf{w}_i$ is an $n \times 1$ vector of standardized DNA methylation measures of the target probe $i$ and $b_i$ is the effect of probe $i$ on the phenotype (fixed effect). Important properties of the MOA model are that the target probe is fitted both as a fixed effect ($b_i$) and a random effect (the $i$th element of the vector $\mathbf{u}$) and effect sizes of probes in the random effect term are assumed to come from a single distribution. The MOMENT approach differs by partitioning probes, on the basis of their statistical association with the phenotype in an initial linear regression, into two random effects with different effect size distributions. This method has been shown to be more robust to potential confounders than MOA but at the cost of slightly reduced power[11]. The MOMENT model is:

$$\mathbf{y} = \mathbf{w}_i b_i + \mathbf{C}\beta + \sum_j \mathbf{W}_j \mathbf{u}_j + \mathbf{e} \qquad (5)$$

where $\mathbf{W}_j$ is an $n \times m_j$ matrix of standardized DNA methylation probe values in the $j$th group, and $m_j$ is the number of DNA methylation sites in the group (excluding the DNA methylation sites <50 Kb from the probe being tested).

Application of MOA and MOMENT to the SGPD data set involved analysis of 263,264 DNA methylation probes in 1638 (851 PD cases, 787 controls) unrelated (estimated pairwise relationship <0.05 calculated using GCTA[44]) European individuals, and for the PEG data set involved analysis of 242,205 probes in 493 European individuals (281 PD cases, 212 controls). Prior to association analyses we regressed DNA methylation measures on known batch effects, which included sentrix row, column and slide in SGPD, and sentrix row and column only in PEG, because slide was significantly associated with disease status in that sample. Sex, predicted age[55], predicted smoking exposure (unpublished work) and predicted CTPs (with the exception of eosinophils) were fitted as fixed effects. We checked strongly associated probes for the presence of common SNPs, using the masking manifests provided by Illumina (https://zwdzwd.github.io/InfiniumAnnotation).

**MWAS meta-analysis of PD**. We performed a MWAS meta-analysis of MOA or MOMENT results for PD using summary data from the SGPD and PEG cohorts, comprising analysis of 229,071 common methylation probes in a total of 1132 PD cases and 999 controls. The analyses were conducted in OSCA, which implements the conventional inverse-weighted-variance analysis assuming independence among cohorts. Novel signals of association were those surpassing the Bonferroni-adjusted significance threshold ($p = 0.05/229, 071 = 2.2 \times 10^{-07}$).

**Summary data-based Mendelian randomization**. The SMR and HEIDI methods[12] were used to identify putative target genes regulated by PD-associated CpG sites, and to examine evidence for causal relationships between DNA methylation, gene expression and PD. Specifically, to infer the regulatory role of epigenome-wide significant probes on PD, we applied the SMR approach to test the hypothesis that DNA methylation is associated with PD through regulation of gene expression[56]. This can be done in the SMR framework by testing the pairwise associations between epigenome-wide CpG probes and PD, epigenome-wide CpG probes and genes within 2 Mb distance in either directions, and between these genes and PD. These analyses involved summary-level SNP data, including Brain-eMeta[57] eQTL data ($n_{eff} = 1194$) obtained from the meta-analysis of GTEx brain[58], CMC[59], and ROSMAP[60] correcting for sample overlap by the MeCS method[57], Brain-mMeta mQTL[57] data ($n_{eff} = 1160$) obtained from the meta-analysis of ROSMAP[60], Hannon et al.[61], and Jaffe et al.[62], and PD GWAS summary data ($n_{eff} = 308,518$) from the PDWBS meta-analysis[63] of 6476 PD cases and 302,042 controls. Consistent with assumptions in the SMR method, we included only genes with at least one cis-eQTL at $P_{eQTL} < 5 \times 10^{-08}$ in Brain-eMeta, leaving 12 genes for analysis.

**DNA methylation-based out-of-sample classification**. In the SGPD cohort, we estimated the aggregated effect of all the probes on disease status using a mixed linear model, which included sex, predicted age, predicted smoking exposure, and predicted CTPs as fixed effects, as implemented in OSCA. Subsequently, the best linear unbiased prediction solutions for the probe effects were calculated and used to generate DNA methylation profile scores for European individuals in the PEG study. Using logistic regression adjusted for sex, predicted age and predicted CTPs, we estimated the proportion of variance in PD status (Nagelkerke $R^2$) in the PEG study captured by the standardized methylation profile scores. In addition, the classification accuracy of the methylation profile scores was evaluated by calculating the AUC.

**Reporting summary**. Further information on research design is available in the Nature Research Reporting Summary linked to this article.

## Data availability

The SGPD methylation data are available from the Gene Expression Omnibus (GEO) (https://www.ncbi.nlm.nih.gov/geo/query/acc.cgi?acc=GSE145361). The SGPD GRM and top two genetic principal components, used to identify unrelated European individuals for inclusion in DNA methylation analyses, are available from (https://cnsgenomics.com/data/vallerga_et_al_2020_nc/). The SGPD MWAS summary statistics are also available from (https://cnsgenomics.com/data/vallerga_et_al_2020_nc/). The PEG methylation data are available from GEO (https://www.ncbi.nlm.nih.gov/geo/query/acc.cgi?acc=GSE111629).

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

## Acknowledgements
We thank all SGPD and PEG participants for donating their time, data, and biological samples, without which this study would not have been possible. This research was supported by the National Health and Medical Research Council (NHMRC: 1078037, 1078901, 1103418, 1107258, 1127440, 1113400), the Australian Research Council (ARC: DP160102400 and FT180100186) and the Mater Foundation. Support also came from ForeFront, a large collaborative research group dedicated to the study of neurodegenerative diseases and funded by the NHMRC (Program Grant 1132524, Dementia Research Team Grant 1095127, NeuroSleep Centre of Research Excellence 1060992) and ARC (Centre of Excellence in Cognition and its Disorders Memory Program CE10001021). S.L. was supported by an NHMRC-ARC Dementia Fellowship (1110414) and G.H. was supported by an NHMRC Fellowship (1079679). The Queensland Parkinson's Project (QPP) was supported by a grant from the Australian National Health and Medical Research Council (1084560) to G.M. The New Zealand Brain Research Institute (NZBRI) cohort was funded by a University of Otago Research Grant, together with financial support from the Jim and Mary Carney Charitable Trust (Whangarei, New Zealand). We thank Allison Miller for processing and handling of NZBRI samples.

## Author contributions
P.M.V. and G.M. conceived the study. G.M., J.F., S.B., G.H., I.H., J.B.K., J.P., T.P., M.K., P.A.S., S.L., T.A., and J.D.-A. collected data and biological samples and performed phenotype data QC. A.K.H. and L.W. generated the SGPD DNA methylation data. C.L.V. and T.Q. performed the primary analyses, with assistance from F.Z., A.F.M., M.F. N., Q.Z., J.Y., N.R.W., P.M.V., and J.G. J.G. and P.M.V. jointly led the project. C.L.V. and J.G. drafted the manuscript and all authors contributed to the final version. S.L., T.A., and J.D.A. contributed equally to the manuscript.

## Competing interests
The authors declare no competing interests.
