## [Peer Review File · Nature Communications]

Reviewers' Comments:

Reviewer #1:

Remarks to the Author:

The paper „Analysis of DNA methylation associates the cystine-glutamate antiporter SLC7A11 with risk of Parkinson's disease „ by Vallerga et al., describes the largest blood cell based EWAS case control study on PD using two larger cohorts.

Overall the paper compiles an impressive amount of 450K data analysis. The author apply a bouquet of ML based statistical methods and combined them with approaches to reduce confounding effects such as CTP. The paper is well structured and well written aiming to convey clear messages focussing on very few "filtered" data points. All computational work and analyses appear to be thoroughly done and are rather well documented.

Major:

While I appreciate the impressive work and deep investigations leading to the identification of CpG positions (one at the SLC7A11), cg06690548, cg26033520, cg16001422, it remains unclear why the real changes in DNA-methylation for the top CpGs are not shown and why none of them has been validated on a smaller cohort using an alternative technological approach (e.g. deep sequencing, ideally on sorted immune cells and whole blood).

Besides focussing on highly significant CpGs, the authors should show and discuss the the list of non-significant (after correction) top EWAS hits (similar to Lunnon et al. 2014 for AD), i.e. their linkage to genes and their potential biological connection and pathways (also mQTLs). Which of the previous top ranking CpGs in PD EWAS studies are present in such top lists or disappear after CTP correction?

The very strong data adjustments, i.e. the normalisation and filtering procedures leading to a reduction of almost half of all data points. This needs to be discussed or rationalized as the adjustment may have led to significant loss of informative disease signals in subcohorts (gender, age).

All functional/biological implication of methylation changes at the three mostly discussed CpGs are exclusively based on public database mining and no link to functional tests are given.

For EWAS cell type proportions (CTP) across multiple samples represent a major confounder in epigenetic analyses – alternatively they might be also seen as an indicator of disease associated cell variation. To reduce confounding imputation (Houseman-method or others) and correction is a very important approach. One surprising outcome of the CTP imputation is that in some samples show almost a 100% neutrophil content while other samples show almost 0% (Fig.1). This is seen for some PD samples but also for some controls. Seeing these extremes one might get the impression that the methods used for samples estimations are quite imprecise. Could alternative measures/methods be taken to check this – since the imputation puts a strong bias on the subsequent analyses, e.g. alternative CTP imputation methods or immunophenotyping assays?

Minor:

The discussion point about BMAA is not new but fueling the interest in a new direction. Are there any (in vitro) experiments/data/studies available that report an impact of BMAA on DNA methylation?

While the paper is written in a comprehensive and well written style the extensive use of abbreviations - sometimes used in long chains- somehow break the reading flow (see e.g. first sentence in table 1). Abbreviations are also being used in figures even if there would be enough space for the full text.

The authors should address the limitations of EWAS run on blood cells for a CNS-related disease and interpret their data with more caution. The apparently non-existing connection to Levodopa treatment should in this respect taken with more caution.

Reviewer #2:

Remarks to the Author:

This manuscript is well written and concise. It represents the first meta-analysis of PD MWAS, which is novel for the field. The analyses seem sound for the most part (see below for specific queries), although in places were fairly descriptive.

1. It would be helpful for the abstract to mention that the MWAS was in blood in line 34 after "two cohorts".
2. Supplementary Figure 1s and 2 - text on axes is too small to read in places - please expand
3. "Supplemetary" misspelt on line 150 page 9
4. Why run both MOA and MOMENT? Why not go with the more conservative MOA? What about other methods for modelling the data; were these considered?
5. The classifier analysis was in the SGPD samples - how good was this in the PEG samples?
6. Similarly, for the proportion of phenotypic variance analysis was the probes that accounted for the 28% and 24% variance in SGPD and PEG, respectively the same probes? If not, how did the probe list from one cohort fare in the other?
7. As genetic data was available for all samples was a disease x genotype mQTL analysis undertaken?
8. What was methylation like in known GWAS genes?
9. The supplmenetary table 3 is the list of probes from SMR. it would be useful to have an excel list of the most significant probes in the two cohorts using the 2 approaches as additional supp tables
10. In the list of probes removed in the methods it does not state that probes known to have common SNPs in the CG or single base extension position were removed. Please clarify whether this was done
11. What can these studies in blood tell us about brain methylation?

Reviewers' comments:

Reviewer #1 (Remarks to the Author):

The paper “Analysis of DNA methylation associates the cystine-glutamate antiporter SLC7A11 with risk of Parkinson’s disease” by Vallerga et al., describes the largest blood cell based EWAS case control study on PD using two larger cohorts. Overall the paper compiles an impressive amount of 450K data analysis. The author apply a bouquet of ML based statistical methods and combined them with approaches to reduce confounding effects such as CTD. The paper is well structured and well written aiming to convey clear messages focussing on very few "filtered" data points. All computational work and analyses appear to be thoroughly done and are rather well documented.

We thank the referee for their time and effort in reviewing our manuscript and for their positive overall assessment of our study.

Major:

While I appreciate the impressive work and deep investigations leading to the identification of CpG positions (one at the SLC7A11), cg06690548, cg26033520, cg16001422, it remains unclear why the real changes in DNA-methylation for the top CpGs are not shown and why none of them has been validated on a smaller cohort using an alternative technological approach (e.g. deep sequencing, ideally on sorted immune cells and whole blood).

We are grateful to the referee for this suggestion. We have now added box plots showing DNA methylation B values for the most strongly associated CpG probes (cg06690548, cg26033520, cg16001422) in PD cases and controls, in SGPD and PEG (see Supplementary Figures 6 & 12 in the revision).

In relation to validation of methylation measures for these probes, we acknowledge that use of an alternative methodology (e.g. targeted bisulphite sequencing) is one approach for studies using Illumina 450K arrays, as in the paper highlighted by the Editor. At the same time, we wish to note that the observation of consistent case-control differences at strongly associated probes in independent datasets is unlikely to occur as a consequence of technical artefacts, particularly after careful QC to remove poorly performing samples and probes. In our study, we performed stringent QC and pre-adjusted probes for batch, cohort, slide, row and column. The two genome-wide significant probes that we report in the meta-analysis exhibited highly consistent case-control differences in the SGPD and PEG cohorts. As these two datasets are completely independent, and as PD cases and controls were randomised onto arrays, it strongly suggests that the associations we report are robust. We also wish to point out that our study is not unusual in taking this analytical approach, since technical validation of top association signals is by no means a universal feature of recently published studies using Illumina 450K and EPIC arrays, including in *Nature Communications* (e.g. Hillary *et al.* 2019, <https://www.nature.com/articles/s41467-019-11177-x>, Cardenas *et al.* 2019, <https://www.nature.com/articles/s41467-019-11058-3>, Kupers *et al.* 2019, <https://www.nature.com/articles/s41467-019-09671-3>).

Besides focussing on highly significant CpGs, the authors should show and discuss the list of non-significant (after correction) top EWAS hits (similar to Lunnon et al. 2014 for AD), i.e. their linkage to genes and their potential biological connection and pathways (also mQTLs).

Which of the previous top ranking CpGs in PD EWAS studies are present in such top lists or disappear after CTP correction?

We thank the Referee for these suggestions, some of which were also made by Referee #2. We have added additional Supplementary Tables containing summary results (including gene names) for CpG probes with $p \leq 1 \times 10^{-3}$ from the MOA and MOMENT analyses in each of SGPD, PEG and the meta-analysis (Supp Tables 3, 4 and 5 respectively). We have also added a further Supplementary Table (#6) with MWAS summary results for CpG probes in known PD genes (e.g. *SCNA*), together with a new sentence in the Results (page 10) noting that:

There was no evidence for strongly associated probes in known PD genes (e.g. SNCA) in either the MOA or MOMENT meta-analyses (Supplementary Table 6).

We trust that the inclusion of these tables – which will enable readers to perform lookups of specific probes or genes – helps to address the referee’s request for further information on non-significant associations. We feel that these tables significantly improve the manuscript, but we would prefer not to expand the discussion text beyond CpGs with genome-wide support in the meta-analysis and/or the SGPD MWAS, since we wish to avoid placing any emphasis on probes that do not survive correction for multiple testing, consistent with modern conventions in human genomics.

In relation to the statistical support in our analyses for previously reported CpGs in PD, we have made several edits to the Results section on page 9, which now reads:

As expected, the effect sizes of CpG probes with $p < 5 \times 10^{-6}$ in the Chuang et al. [8] PEG analysis in which methylation was adjusted for blood CTPs were highly correlated with those in our analysis of the PEG cohort ($n = 10$ probes, Pearson’s correlation = 0.93, $p = 8.9 \times 10^{-5}$, Supplementary Figure 10), as were the effect sizes of genome-wide significant probes reported by Chuang et al. in their analysis without adjustment for blood CTPs ($N = 78$ shared probes, Pearson’s correlation = 0.77, $p < 2.2 \times 10^{-16}$, Supplementary Figure 11). However, none of the 78 shared probes was genome-wide significant in our analysis of the PEG data set, and nor did any single shared probe replicate in the SGPD data ($p < 0.05/78$).

The very strong data adjustments, i.e. the normalisation and filtering procedures leading to a reduction of almost half of all data points. This needs to be discussed or rationalized as the adjustment may have led to significant loss of informative disease signals in subcohorts (gender, age).

The referee is correct that our filtering procedures removed approximately half of the probes in each cohort. We have summarised the number of probes removed from the SGPD and PEG datasets at each stage in our filtering procedure in the Table below:

Filtering step	SGPD		PEG	
	Probes removed	Probes remaining	Probes removed	Probes remaining
Cross-reactive probes	29218	456019	29198	455739
X/Y chromosome probes	10711	445308	10742	444997
SD < 0.02 probes	182044	263264	202792	242205

As this shows, the majority (i.e. >80%) of the probes removed in each cohort (i.e. 182K of 222K probes filtered in SGPD & 203K of 242K filtered in PEG) were lowly variable probes (SD < 0.02), for which there is limited power to detect a case-control difference in methylation levels. The

justification for removing these probes is that it increases our statistical power by reducing the multiple testing burden.

To confirm that our filtering strategy did not lead us to inadvertently miss any important association signals, we repeated our analyses with the low SD probes included. As shown in the QQ plots below, no single low SD probe was found to be genome-wide significant (indicated by “+”) in either the MOA or MOMENT MWAS meta-analysis of SGPD and PEG (top = MOA, bottom = MOMENT, left = probes included in the original analyses ($SD \geq 0.02$), right = lowly variable probes ($SD < 0.02$)):

Conversely, although the motivation for filtering low SD probes was to increase statistical power, we wish to note that the two genome-wide significant probes in the MOA meta-analysis (cg06690548, cg26033520) would nonetheless surpass genome-wide significance in an analysis correcting for all probes.

We hope that this additional information helps to reassure the Referee that our QC and analysis strategy is well reasoned, robust and designed to maximise the likelihood of identifying true associations between DNA methylation and PD, while at the same time minimising the possibility of declaring false positive associations.

All functional/biological implication of methylation changes at the three mostly discussed CpGs are exclusively based on public database mining and no link to functional tests are given.

We acknowledge the Referee's point, but the purpose of this study was to identify statistical associations between DNA methylation and PD status, and to the extent possible, to use the SMR method and publicly available brain eQTL and mQTL data to (1) link associated CpGs with target genes, (2) evaluate evidence for causal relationships between SNPs, DNA methylation, gene expression and PD. The scope of our study does not extend to functional validation of identified signals. However, we provide multiple candidate targets for the PD functional genomics community to pursue.

For EWAS cell type proportions (CTP) across multiple samples represent a major confounder in epigenetic analyses – alternatively they might be also seen as an indicator of disease associated cell variation. To reduce confounding imputation (Houseman-method or others) and correction is a very important approach. One surprising outcome of the CTP imputation is that in some samples show almost a 100% neutrophil content while other samples show almost 0% (Fig.1). This is seen for some PD samples but also for some controls. Seeing these extremes one might get the impression that the methods used for samples estimations are quite imprecise. Could alternative measures/methods be taken to check this – since the imputation puts a strong bias on the subsequent analyses, e.g. alternative CTP imputation methods or immunophenotyping assays?

The referee is correct that CTPs represent an important confounder in analyses of DNA methylation, and that some samples in our data showed very high (or low) estimated neutrophil proportions. However, we are confident that any potential imprecision in estimated CTPs obtained using the Houseman *et al.* (2012) method – now cited >1250 times – did not adversely influence our results, and we are pleased to have the opportunity to explain why.

The mixed model-based analysis methods used in our study (MOA and MOMENT; see Zhang et al. 2019, *OSCA: a tool for omic-data-based complex trait analysis. Genome Biology* **20**: 107) fit one or more random effects comprising all probes (or distal probes) when testing for association with a specific probe. The rationale for (and major benefit of) this approach is that it accounts for both known (e.g. sex, age, smoking, CTPs) and unknown (or unmeasured) confounders. In our analyses, we conservatively fitted predicted CTPs, sex, predicted age and predicted smoking as covariates (i.e. consistent with accepted norms in the field) in addition to the random effect(s). However, when we repeat our analyses without fitting CTPs, sex, age or smoking, the beta estimates are highly correlated with those from our original analyses – see plots below (left panel = SGPD, right panel = meta-analysis):

Indeed, the beta correlations are especially high for the probes that were most strongly associated ($p < 1 \times 10^{-4}$) with PD in the original analyses (left panel = SGPD, right panel = meta-analysis):

Moreover, we observe the same strong effect size correlations when we compare our original analyses with models in which we adjust for sex, age and smoking, but not CTPs, as shown in the plots below (left panel = SGPD, right panel = meta-analysis):

In all of these repeat analyses without adjustment for covariates, the probes reported as genome-wide significant in our original analyses (i.e. cg06690548, cg26033520, cg16001422) remained genome-wide significant.

As the results from models without adjustment for CTPs (or indeed sex, age and smoking) are so consistent with our reported results in which we (conservatively) adjust for these confounders, it strongly suggests that our reported findings are not due to any potential imprecision in the predicted CTPs. These results also highlight that the MOA and MOMENT methods effectively control for known confounders of DNA methylation, such as CTPs, sex, age and smoking, and so therefore likely also account for unmeasured confounders. This is a major advantage of the mixed linear model-based methods used in our study, as compared to conventional analysis methods such as linear or logistic regression, which can only correct for known and measured confounders.

Minor:

The discussion point about BMAA is not new but fueling the interest in a new direction. Are there any (in vitro) experiments/data/studies available that report an impact of BMAA on DNA methylation?

We are not aware of any studies that demonstrate an effect of BMAA exposure on DNA methylation. We anticipate that our study will stimulate such investigations.

While the paper is written in a comprehensive and well written style the extensive use of abbreviations - sometimes used in long chains- somehow break the reading flow (see e.g. first sentence in table 1). Abbreviations are also being used in figures even if there would be enough space for the full text.

We have carefully revised the manuscript, including the Table and Figure captions, to limit the number of abbreviations. We now retain only those that appear very frequently in the text (e.g. PD, SGPD, PEG, MOA, MOMENT, BMAA, MWAS, CTPs), or that are unwieldy when written in full (e.g. OSCA = OmicsS-data-based Complex trait Analysis; AUC = area under the receiver operator characteristic curve). We have removed all other abbreviations mentioned fewer than ~10 times in the text (e.g. BLUP, GSH, MLM, MPS, ROC, SNpc) and now include a list of all remaining abbreviations as a Supplementary Note. We trust that these changes have improved the style and flow of the manuscript.

The authors should address the limitations of EWAS run on blood cells for a CNS-related disease and interpret their data with more caution.

We thank the Referee for this suggestion and have added the following text to the limitations section of the Discussion (page 19):

We also acknowledge that analysis of whole blood-based DNA is a limitation, given that PD is a disorder of the central and peripheral nervous systems. Analysis of DNA methylation in post-mortem brain tissues from PD cases and controls should be a future priority for the field.

We also refer the Referee to our response to Referee #2, point #11 (see below).

The apparently non-existing connection to Levodopa treatment should in this respect taken with more caution.

We thank the referee for bringing this additional caveat to our attention. We agree and have edited the Discussion to be more circumspect in our interpretation of the lack of an association between cg06690548 methylation and LEDD exposure, given that the measurements are in blood. The relevant section of the Discussion (page 14-15) now reads:

We found no evidence for an association of cg06690548 methylation with PD medication dosage ($\beta = 0.50$ s.e. = 0.37, $p = 0.17$; unpubl. data), although we cannot rule out the possibility of a brain-specific association.

Reviewer #2 (Remarks to the Author):

This manuscript is well written and concise. It represents the first meta-analysis of PD MWAS, which is novel for the field. The analyses seem sound for the most part (see below for specific queries), although in places were fairly descriptive.

We thank the Referee for their time and positive assessment of our manuscript.

1. It would be helpful for the abstract to mention that the MWAS was in blood in line 34 after "two cohorts".

This is an excellent suggestion from the Referee and we have edited the abstract accordingly. The second sentence of the abstract now reads:

We present the largest blood-based methylome-wide association study (MWAS) of Parkinson's disease (PD) to date, involving meta-analysis of 229K CpG probes in 1,132 cases and 999 controls from two independent cohorts.

2. Supplementary Figure 1s and 2 - text on axes is too small to read in places - please expand

We thank the referee for bringing this to our attention. We have increased the font size for the axes labels on both of these Supplementary Figures.

3. "Supplemetary" misspelt on line 150 page 9

Thank you for identifying this typo. This has been corrected.

4. Why run both MOA and MOMENT? Why not go with the more conservative MOA? What about other methods for modelling the data; were these considered?

MOA and MOMENT are newly developed and published mixed linear model-based methods for assessing associations between a trait and DNA methylation (see Zhang et al. 2019, *OSCA: a tool for omic-data-based complex trait analysis. Genome Biology* **20**: 107 – reference 11 in the main text of the revised manuscript). We selected these methods because they have been shown by simulation to have a lower false positive rate and to be more robust and conservative than conventional analysis methods such as linear or logistic regression. A feature of both MOA and MOMENT is that when testing for association with a specific probe they fit one or more random effects comprising all other probes (or distal probes) to account for unobserved confounders. We report results from both MOA and MOMENT because whereas MOMENT has a lower false positive rate, MOA is slightly more powerful. This strategy means we maximise our power for discovery using MOA but can have high confidence in the identified associations because we observed consistent results using MOMENT (i.e. indicating that the MOA associations are unlikely to be due to confounding). We anticipate that these mixed linear model-based methods for MWAS will be widely adopted in the field due to these advantages.

5. The classifier analysis was in the SGPD samples - how good was this in the PEG samples?

In the classifier analysis, we used the SGPD data to estimate effect sizes for each CpG methylation probe and then used these estimates to generate methylation profile scores for each individual in the PEG sample. In other words, SGPD was the discovery (“training”) cohort and PEG was the independent target (“test”) cohort. The results in Figure 4 show the performance of the SGPD-based methylation classifier in the PEG sample. The use of independent discovery and target datasets is acknowledged as best practice in analyses of this type. We did not perform the reciprocal analysis, using PEG as the training data and SGPD as the test data, because it is optimal in classification analyses to maximise the size of the training dataset.

6. Similarly, for the proportion of phenotypic variance analysis was the probes that accounted for the 28% and 24% variance in SGPD and PEG, respectively the same probes? If not, how did the probe list from one cohort fare in the other?

The estimates of the proportion of phenotypic variance in PD associated with DNA methylation were based on all the DNA methylation probes in the respective cohorts (i.e. all the data), rather than a subset of probes. The estimates are highly consistent in the two samples (SGPD = 0.28, s.e.= 0.05 and PEG = 0.24, s.e.= 0.12).

7. As genetic data was available for all samples was a disease x genotype mQTL analysis undertaken?

We thank the Referee for this suggestion, but detailed mQTL analyses are the subject of other manuscripts, both in the SGPD data and PEG.

8. What was methylation like in known GWAS genes?

Thank you for prompting us to address this question. Although the specific genes underlying many GWAS associations in PD remain unknown, there are numerous well-established risk genes for PD (e.g. *SNCA*). We have now added a new Supplementary Table (#6) and an additional sentence to the Results, in which we note that probes in well-known PD genes show no evidence for significant DNA methylation differences between PD cases and controls. The relevant section of the Results (page 10) reads:

There was no evidence for strongly associated probes in known PD genes (e.g. SNCA) in either the MOA or MOMENT meta-analyses (Supplementary Table 6).

9. The supplementary table 3 is the list of probes from SMR. it would be useful to have an excel list of the most significant probes in the two cohorts using the 2 approaches as additional supp tables

This is another helpful suggestion. We have now added additional Supplementary Tables containing summary results for probes with $p \leq 1 \times 10^{-03}$ in each cohort, for each analysis approach (MOA and MOMENT). Results for SGPD, PEG and the meta-analysis are summarized in Supplementary Tables 3, 4 and 5, respectively. Each file contains separate worksheets for the MOA and MOMENT analyses.

10. In the list of probes removed in the methods it does not state that probes known to have common SNPs in the CG or single base extension position were removed. Please clarify whether this was done

This is an excellent question and we should have been clearer about our procedures for dealing with common SNPs in the 3' 5bp sub-sequence of probes. We chose not to remove probes with common SNPs up-front, since we know that many assays with SNPs at the CpG site are functional (e.g. see Chundru *et al.* 2019, Genetics 212: 577–586, doi: [10.1534/genetics.118.301861](https://doi.org/10.1534/genetics.118.301861)) and we did not want to miss true associations. Instead, we used the masking manifests for the Illumina 450K array (<https://zwdzwd.github.io/InfiniumAnnotation>) to check if any of the strongly associated probes harboured a common (or rare) SNP in the 3' 5bp sub-sequence. Details of these masks are now included in Supplementary Tables 3, 4 and 5 for CpG probes with $p \leq 1e-03$ in the MOA and MOMENT analyses of SGPD, PEG and the meta-analysis (respectively). We can confirm that no common SNPs are present in either of the two CpG probes surpassing genome-wide significance in the MOA meta-analysis of SGPD and PEG (i.e. cg06690548, cg26033520). A rare SNP is present in cg26033520, but this has frequency of only 3.2×10^{-05} (gnomAD v2.1.1) and thus is highly unlikely to have been observed in our cohort or to have influenced our results. We have now added the following text to the Results (page 10):

Neither of the two epigenome-wide significant probes in the MOA meta-analysis harboured a common SNP in the 3' 5bp-subsequence of the probe that could have influenced our results.

And the following text to the Methods (page 29):

We checked strongly associated probes for the presence of common SNPs, using the masking manifests provided by Illumina (<https://zwdzwd.github.io/InfiniumAnnotation>).

11. What can these studies in blood tell us about brain methylation?

Referee #1 also raised this important limitation of our study, which we have now acknowledged in the Discussion (see response to Referee #1, second to last comment). Notwithstanding that analysis of DNA methylation in post-mortem brain tissues from PD cases and controls should be a high priority for future research, it is important to note that such studies also have limitations. Brain tissue is inaccessible and therefore of limited use for the identification of biomarkers relevant to diagnosis, disease progression and treatment response. Available post-mortem brain tissue collections are also modest in size, and thus brain-based methylation studies are likely to be underpowered in comparison to blood-based analyses. There is also evidence that the genetic control of DNA methylation in blood and brain is strongly correlated (see Qi *et al.* 2018, Identifying gene targets for brain-related traits using transcriptomic and methylomic data from blood. *Nature Communications*, 2018, **9**: 2282), suggesting that MWAS of blood-based DNA for PD and other neurological disorders may be relevant not only for biomarker driven studies, but also those primarily focused on understanding disease etiology.

Reviewers' Comments:

Reviewer #1:

Remarks to the Author:

Excellent and qualified answers to my questions - no further comment.

Reviewer #2:

Remarks to the Author:

I am satisfied that the authors have adequately addressed all of my original comments (reviewer 2). One further point to address would be clearer legends and column heads for the tables that have been added in this revision in response to my comments (supp tables 3-6). In all of these the legends need further detail and many of the column heads are not clear what that column refers to. Finally, for the testing of differential methylation in known PD variants, this could have been done more systematically than just looking at the raw p values in all probes in each gene - if the p values were combined within genes, or even LD blocks, was there any enrichment looking regionally?

Reviewer #1 (Remarks to the Author):

Excellent and qualified answers to my questions - no further comment.

We thank the Referee for taking the time to assess our revised manuscript, and for their positive appraisal of our responses.

Reviewer #2 (Remarks to the Author):

I am satisfied that the authors have adequately addressed all of my original comments (reviewer 2). One further point to address would be clearer legends and column heads for the tables that have been added in this revision in response to my comments (supp tables 3-6). In all of these the legends need further detail and many of the column heads are not clear what that column refers to. Finally, for the testing of differential methylation in known PD variants, this could have been done more systematically than just looking at the raw p values in all probes in each gene - if the p values were combined within genes, or even LD blocks, was there any enrichment looking regionally?

We thank the Referee for their time and effort in assessing our revised manuscript. We are pleased to have satisfactorily addressed their original comments.

We are grateful to the Referee for bringing our attention to the legends and column names for Supp Tables 3-6 (now renamed as Supplementary Data 1-4). We have edited the legends and column heads to provide additional detail, and we trust these data files are now clear.

In relation to the Referee's query about differential methylation testing in known PD genes, we acknowledge that gene- and/or region-based testing are alternative approaches to identifying methylation associations, and that in some cases this may reveal case-control differences not apparent when testing individual probes. However, the most strongly associated CpG probes across the known PD genes had p-values four to five orders of magnitude less significant than the genome-wide significance threshold. We maintain that this observation is notable, relative to strong associations in other genes. Our results imply that there is limited evidence for association of DNA methylation variation with probes in any established PD gene, although it does not preclude the identification of associations in these genes in future, larger MWAS of PD. Given the weak signal from analysis of individual probes in these genes in our study, and the necessity to correct for ~20K genes in gene-based analyses of DNA methylation, we respectfully argue that gene- and/or region-based testing of known genes is unlikely to reveal any significant methylation associations in known PD genes.